# Shape and Material from Sound

**Zhoutong Zhang**
MIT

**Qiujia Li**
University of Cambridge

**Zhengjia Huang**
ShanghaiTech University

**Jiajun Wu**
MIT

**Joshua B. Tenenbaum**
MIT

**William T. Freeman**
MIT, Google Research

## Abstract

Hearing an object falling onto the ground, humans can recover rich information including its rough shape, material, and falling height. In this paper, we build machines to approximate such competency. We first mimic human knowledge of the physical world by building an efficient, physics-based simulation engine. Then, we present an analysis-by-synthesis approach to infer properties of the falling object. We further accelerate the process by learning a mapping from a sound wave to object properties, and using the predicted values to initialize the inference. This mapping can be viewed as an approximation of human commonsense learned from past experience. Our model performs well on both synthetic audio clips and real recordings without requiring any annotated data. We conduct behavior studies to compare human responses with ours on estimating object shape, material, and falling height from sound. Our model achieves near-human performance.

## 1 Introduction

From a short audio clip of interacting objects, humans can recover the number of objects involved, as well as their materials and surface smoothness [Zwicker and Fastl, 2013, Kunkler-Peck and Turvey, 2000, Siegel et al., 2014]. How does our cognitive system recover so much content from so little? What is the role of past experience in understanding auditory data?

For physical scene understanding from visual input, recent behavioral and computational studies suggest that human judgments can be well explained as approximate, probabilistic simulations of a mental physics engine [Battaglia et al., 2013, Sanborn et al., 2013]. These studies suggest that the brain encodes rich, but noisy, knowledge of physical properties of objects and basic laws of physical interactions between objects. To understand, reason, and predict about a physical scene, humans seem to rely on simulations from this mental physics engine.

In this paper, we develop a computational system to interpret audio clips of falling objects, inspired by the idea that humans may use a physics engine as part of a generative model to understand the physical world. Our generative model has three components. The first is a object representation that includes its 3D shape, position in space, and physical properties such as mass, Young's modulus, Rayleigh damping coefficients, and restitution. We aim to infer all these attributes from auditory inputs.

The second component is an efficient, physics-based audio synthesis engine. Given an initial scene setup and object properties, the engine simulates the object's motion and generates its trajectory using rigid body physics. It also produces the corresponding *collision profile* — when, where, and how collisions happen. The object's trajectory and collision profile are then combined with its pre-computed sound statistics to generate the sound it makes during the physical event. With this efficient forward model, we can then infer object properties using analysis-by-synthesis; for each audio clip, we want to find a set of latent variables that best reproduce it.

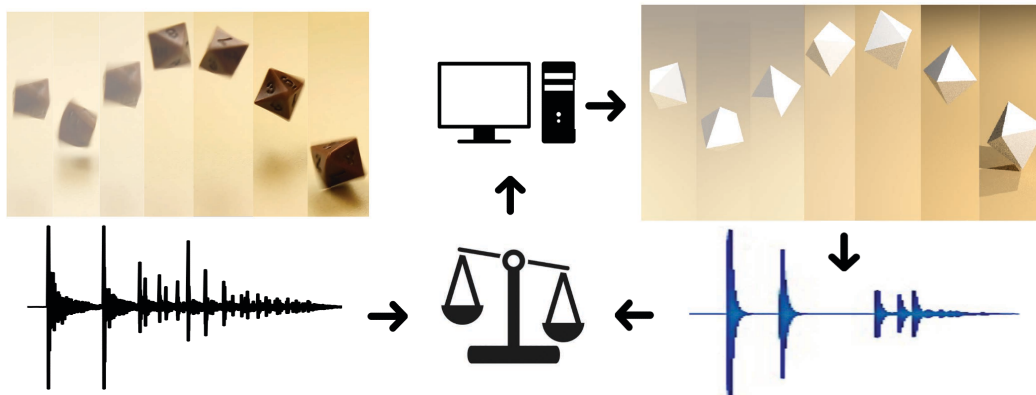

Figure 1: Given an audio of a single object falling, we utilize our generative model to infer latent variables that could best reproduce the sound.

The third component of the model is therefore a likelihood function that measures the perceptual distance between two sounds. Designing such a likelihood function is typically challenging; however, we observe that features like spectrogram are effective when latent variables have limited degrees of freedom. This motivates us to infer latent variables via methods like Gibbs sampling, where we focus on approximating the conditional probability of a single variable given the others.

The inference procedure can be further accelerated with a self-supervised learning paradigm inspired by the wake/sleep phases in Helmholtz machines [Dayan et al., 1995]. We train a deep neural network as the recognition model to regress object properties from sound, where training data are generated using our inference algorithm. Then, for any future audio clip, the output of the recognition model can be used as a good initialization for the sampling algorithm to converge faster.

We evaluate our models on a range of perception tasks: inferring object shape, material, and initial height from sound. We also collect human responses for each task and compare them with model estimates. Our results indicate that first, humans are quite successful in these tasks; second, our model not only closely matches human successes, but also makes similar errors as humans do. For these quantitative evaluations, we have mostly used synthetic data, where ground truth labels are available. We further evaluate the model on recordings to demonstrate that it also performs well on real-world audios.

We make three contributions in this paper. First, we propose a novel model for estimating physical object properties from auditory inputs by incorporating the feedback of a physics engine and an audio engine into the inference process. Second, we incorporate a deep recognition network with the generative model for more efficient inference. Third, we evaluate our model and compare it to humans on a variety of judgment tasks, and demonstrate the correlation between human responses and model estimates.

## 2    Related Work

**Human visual and auditory perception**    Psychoacoustics researchers have explored how humans can infer object properties, including shape, material and size, from audio in the past decades [Zwicker and Fastl, 2013, Kunkler-Peck and Turvey, 2000, Rocchesso and Fontana, 2003, Klatzky et al., 2000, Siegel et al., 2014]. Recently, McDermott et al. [2013] proposed compact sound representations that capture semantic information and are informative of human auditory perception.

**Sound simulation**    Our sound synthesis engine builds upon and extends existing sound simulation systems in computer graphics and computer vision [O'Brien et al., 2001, 2002, James et al., 2006, Bonneel et al., 2008, Van den Doel and Pai, 1998, Zhang et al., 2017]. Van den Doel and Pai [1998] simulated object vibration using the finite element method and approximated the vibrating object as a single point source. O'Brien et al. [2001, 2002] used the Rayleigh method to approximate wave equation solutions for better synthesis quality. James et al. [2006] proposed to solve Helmholtz

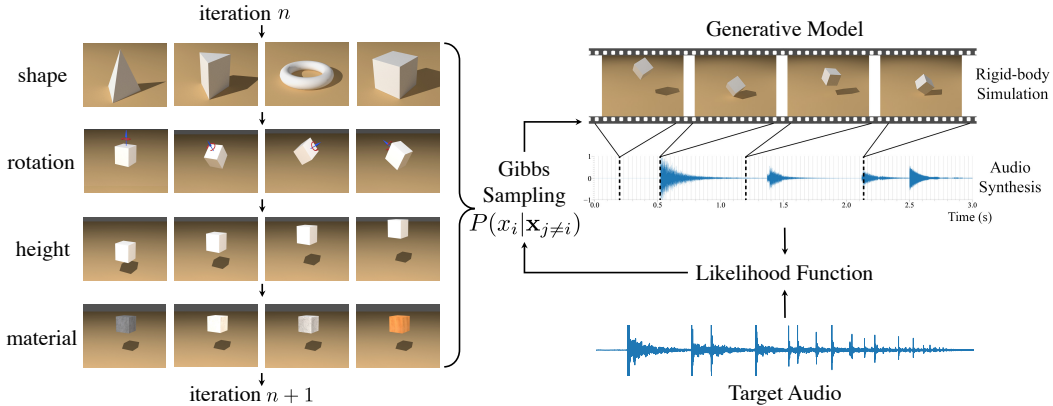

Figure 2: Our inference pipeline. We use Gibbs sampling over the latent variables. The conditional probability is approximated using the likelihood between reconstructed sound and the input sound.

equations using the Boundary Element Method, where each object's vibration mode is approximated by a set of vibrating points. Recently, Zhang et al. [2017] built a framework for synthesizing large-scale audio-visual data. In this paper, we accelerate the framework by Zhang et al. [2017] to achieve near real-time rendering, and explore learning object representations from sound with the synthesis engine in the loop.

**Physical Object Perception**   There has been a growing interest in understanding physical object properties, like mass and friction, from visual input or scene dynamics [Chang et al., 2017, Battaglia et al., 2016, Wu et al., 2015, 2016, 2017]. Much of the existing research has focused on inferring object properties from visual data. Recently, researchers have begun to explore learning object representations from sound. Owens et al. [2016a] attempted to infer material properties from audio, focusing on the scenario of hitting objects with a drumstick. Owens et al. [2016b] further demonstrated audio signals can be used as supervision on learning object concepts from visual data, and Aytar et al. [2016] proposed to learn sound representations from corresponding video frames. Zhang et al. [2017] discussed the complementary role of auditory and visual data in recovering both geometric and physical object properties. In this paper, we learn physical object representations from audio through a combination of powerful deep recognition models and analysis-by-synthesis inference methods.

**Analysis-by-synthesis**   Our framework also relates to the field of analysis-by-synthesis, or generative models with data-driven proposals [Yuille and Kersten, 2006, Zhu and Mumford, 2007, Wu et al., 2015], as we are incorporating a graphics engine as a black-box synthesizer. Unlike earlier methods that focus mostly on explaining visual data, our work aims to infer latent parameters from auditory data. Please refer to Bever and Poeppel [2010] for a review of analysis-by-synthesis methods.

# 3   An Efficient, Physics-Based Audio Engine

At the core of our inference pipeline is an efficient audio synthesis engine. In this section, we first give a brief overview of existing synthesis engines, and then present our technical innovations on accelerating them for real-time rendering in our inference algorithm.

## 3.1   Audio Synthesis Engine

Audio synthesis engines generate realistic sound by simulating physics. First, rigid body simulation produces the interaction between an object and the environment, where Newton's laws dictate the object's motion and collisions over time. Each collision causes the object to vibrate in certain patterns, changing the air pressure around its surface. These vibrations propagate in air to the recorder and create the sound of this physical process.

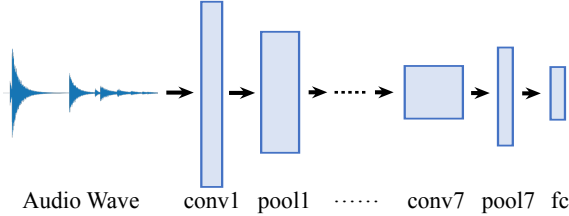

Figure 3: Our 1D deep convolutional network. Its architecture follows that in Aytar et al. [2016], where raw audio waves are forwarded through consecutive conv-pool layers, and then passed to a fully connected layer to produce output.

| Settings | Time ($s$) |
|---|---|
| Original algorithm | 30.4 |
| Amplitude cutoff | 24.5 |
| Principal modes | 12.7 |
| Multi-threading | 1.5 |
| **All** | **0.8** |

Table 1: Acceleration break down of each technique we adopted. Timing is evaluated by synthesizing an audio with 200 collisions. The last row reports the final timing after adopting all techniques.

**Rigid Body Simulation**   Given an object's 3D position and orientation, and its mass and restitution, a physics engine can simulate the physical processes and output the object's position, orientation, and collision information over time. Our implementation uses an open-source physics engine, Bullet [Coumans, 2010]. We use a time step of 1/300 second to ensure simulation accuracy. At each time step, we record the 3D pose and position of the object, as well as the location, magnitude, and direction of collisions. The sound made by the object can then be approximated by accumulating sounds caused by those discrete impulse collisions on its surface.

**Audio Synthesis**   The audio synthesis procedure is built upon previous work on simulating realistic sounds [James et al., 2006, Bonneel et al., 2008, O'Brien et al., 2001]. To facilitate fast synthesis, this process is decomposed into two modules, one offline and one online. The offline part first uses finite element methods (FEM) to obtain the object's vibration modes, which depend on the shape and Young's modulus of the object. These vibration modes are then used as Neumann boundary conditions of the Helmholtz equation, which can be solved using boundary element methods (BEM). We use the techniques proposed by James et al. [2006] to approximate the solution by modeling the pressure fields with a sparse set of vibrating points. Note that the computation above only depends on object's intrinsic properties such as shape and Young's modulus, but not on the extrinsics such as its position and velocity. This allows us to pre-compute a number of shape-modulus configurations before simulation; only minimal computation is needed during the online simulation.

The online part of the audio engine loads pre-computed approximations and decomposes impulses on the surface mesh of the object into its modal bases. At the observation point, the engine measures the pressure changes induced by vibrations in each mode, and sums them up to produce the simulated sound. An evaluation of the fidelity of these simulations can be found in Zhang et al. [2017].

### 3.2    Accelerating Audio Synthesis

Analysis-by-synthesis inference requires the audio engine to be highly efficient; however, a straight-forward implementation of the above simulation procedure would be computationally expensive. We therefore present technical innovations to accelerate the computation to near real-time.

First, we select the most significant modes excited by each impulse until their total energy reaches 90% of the energy of the impulse. Ignoring sound components generated by the less significant modes reduces the computational time by about 50%. Second, we stop the synthesis process if the amplitude of the damped sound goes below a certain threshold, since it is unlikely to be heard. Third, we parallelize the synthesis process by tackling collisions separately, so that each can computed on an independent thread. We then integrate them into a shared buffer to generate the final audio according to their timestamps. The effect of acceleration is shown in Table 1. Online sound synthesis only contains variables that are fully decoupled from the offline stage, which enables us to freely manipulate other variables with little computational cost during simulation.

### 3.3    Generating Stimuli

Because real audio recordings with rich labels are hard to acquire, we synthesize random audio clips using our physics-based simulation to evaluate our models. Specifically, we focus on a single

| Variable | Range | C/T | Variable | Range | C/T |
|---|---|---|---|---|---|
| Primitive shape ($s$) | 14 classes | D | Specific modulus ($E/\rho$) | $[1, 30] \times 10^6$ | D |
| Height ($z$) | $[1, 2]$ | C | Restitution ($e$) | $[0.6, 0.9]$ | C |
| Rotation axis ($i, j, k$) | $S^2$ | C | Rotation angle ($w$) | $[-\pi, \pi)$ | C |
| Rayleigh damping ($\alpha$) | $10^{[-8,-5]}$ | C | Rayleigh damping ($\beta$) | $2^{[0,5]}$ | C |

Table 2: Variables in our generative model, where the C/T column indicates whether sampling takes place in continuous (C) or discrete (D) domain, and values inside parentheses are the range we uniformly sampled from. Rotation is defined in quaternions.

scenario — shape primitives falling onto the ground. We first construct an audio dataset that includes 14 primitives (some shown in Table 2), each with 10 different specific moduli (defined as Young's modulus over density). After pre-computing their space-modulus configurations, we can generate synthetic audio clips in a near-real-time fashion. Because the process of objects falling onto the ground is relatively fast, we set the total simulation time of each scenario to 3 seconds. Details of our setup can be found in Table 2.

# 4   Inference

In this section, we investigate four models for inferring object properties, each corresponding to a different training condition. Inspired by how humans can infer scene information using a mental physics engine [Battaglia et al., 2013, Sanborn et al., 2013], we start from an unsupervised model where the input is only one single test case with no annotation. We adopt Gibbs sampling over latent variables to find the combination that best reproduces the given audio.

We then extend the model to include a deep neural network, analogous to what humans may learn from their past experience. The network is trained using labels inferred by the unsupervised model. During inference, the sampling algorithm uses the network prediction as the initialization. This self-supervised learning paradigm speeds-up convergence.

We also investigate a third case, when labels can be acquired but are extremely coarse. We first train a recognition model with weak labels, then randomly pick candidates from those labels as an initialization for our analysis-by-synthesis inference.

Lastly, to understand performance limits, we train a deep neural network with fully labeled data that yields the upper-bound performance.

## 4.1   Models

**Unsupervised**   Given an audio clip $S$, we would like to recover the latent variables $\mathbf{x}$ to make the reproduced sound $g(\mathbf{x})$ most similar to $S$. Let $L(\cdot, \cdot)$ be a likelihood function that measures the perceptual distance between two sounds, then our goal is to maximize $L(g(\mathbf{x}), S)$. We denote $L(g(\mathbf{x}), S)$ as $p(\mathbf{x})$ for brevity. In order to find $\mathbf{x}$ that maximizes $p(\mathbf{x})$, $p(\mathbf{x})$ can be treated as an distribution $\hat{p}(\mathbf{x})$ scaled by an unknown partition function $Z$. Since we do not have an exact form for $p(\cdot)$, nor $\hat{p}(\mathbf{x})$, we apply Gibbs sampling to draw samples from $p(\mathbf{x})$. Specifically, at sweep round $t$, we update each variable $\mathbf{x}_i$ by drawing samples from

$$\hat{p}(x_i | x_1^t, x_2^t, ... x_{i-1}^t, x_{i+1}^{t-1}, ... x_n^{t-1}). \tag{1}$$

Such conditional probabilities are straightforward to approximate. For example, to sample Young's modulus conditioned on other variables, we can use the spectrogram as a feature and measure the $l_2$ distance between the spectrograms of two sounds, because Young's modulus will only affect the frequency at each collision. Indeed, we can use the spectrogram as features for all variables except height. Since the height can be inferred from the time of the first collision, a simple likelihood function can be designed as measuring the time difference between the first impact in two sounds. Note that this is only an approximate measure: object's shape and orientation also affect, although only slightly, the time of first impact.

To sample from the conditional probabilities, we adopt the Metropolis–Hastings algorithm, where samples are drawn from a Gaussian distribution and are accepted by flipping a biased coin according to its likelihood compared with the previous sample. Specifically, we calculate the $l_2$ distance $d^t$ in feature space between $g(\mathbf{x}^t)$ and $S$. For a new sample $\mathbf{x}^{t+1}$, we also calculate the $l_2$ distance $d^{t+1}$ in feature space between $g(\mathbf{x}^{t+1})$ and $S$. The new sample is accepted if $d^{t+1}$ is smaller than $d^t$; otherwise, $\mathbf{x}^{t+1}$ is accepted with probability $\exp(-(d^{t+1} - d^t)/T)$, where $T$ is a time varying function inspired by simulated annealing algorithm. In our implementation, $T$ is set as a quadratic function of the current MCMC sweep number $t$.

**Self-supervised Learning**   To accelerate the above sampling process, we propose a self-supervised model, which is analogous to a Helmholtz machine trained by the wake-sleep algorithm. We first train a deep neural network, whose labels are generated by the unsupervised inference model suggested above for a limited number of iterations. For a new audio clip, our self-supervised model uses the result from the neural network as an initialization, and then runs our analysis-by-synthesis algorithm to refine the inference. By making use of the past experiences which trained the network, the sampling process starts from a better position and requires fewer iterations to converge than the unsupervised model.

**Weakly-supervised Learning**   We further investigate the case where weak supervision might be helpful for accelerating the inference process. Since the latent variables we aim to recover are hard to obtain in real world settings, it is more realistic to assume that we could acquire very coarse labels, such as the type of material, rough attributes of the object's shape, the height of the fall, *etc*. Based on such assumptions, we coarsen ground truth labels for all variables. For our primitive shapes, three attributes are defined, namely "with edge," "with curved surface," and "pointy." For material parameters, *i.e.*, specific modulus, Rayleigh damping coefficients and restitution, they are mapped to steel, ceramic, polystyrene and wood by finding the nearest neighbor to those real material parameters. Height is divided into "low" and "high" categories. A deep convolutional neural network is trained on our synthesized dataset with coarse labels. As shown in Figure 4, even trained using coarse labels, our network learns features very similar to the ones learned by the fully supervised network. To go beyond coarse labels, the unsupervised model is applied using the initialization suggested by the neural network.

**Fully-supervised Learning**   To explore the performance upper bound of the inference tasks, we train an oracle model with ground truth labels. To visualize the abstraction and characteristic features learned by the oracle model, we plot the inputs that maximally activate some hidden units in the last layer of the network. Figure 4 illustrates some of the most interesting waveforms. A selection of them learned to recognize specific temporal patterns, and others were sensitive to specific frequencies. Similar patterns were found in the weakly and fully supervised models.

## 4.2   Contrasting Model Performance

We evaluate how well our model performs under different settings, studying how past experience or coarse labeling can improve the unsupervised results. We first present the implementation details of all four models, then compare their results on all inference tasks.

**Sampling Setup**   We perform 80 sweeps of MCMC sampling over all the 7 latent variables; for every sweep, each variable is sampled twice. Shape, specific modulus and rotation are sampled by uniform distributions across their corresponding dimensions. For other continuous variables, we define an auxiliary Gaussian variable $x_i \sim \mathcal{N}(\mu_i, \sigma_i^2)$ for sampling, where the mean $\mu_i$ is based on the current state. To evaluate the likelihood function between the input and the sampled audio (both with sample rate of 44.1k), we compute the spectrogram of the signal using a Tukey window of length 5,000 with a 2,000 sample overlap. For each window, a 10,000 point Fourier transform is applied.

**Deep Learning Setup**   Our fully supervised and self-supervised recognition models use the architecture of SoundNet-8 [Aytar et al., 2016] as Figure 3, which takes an arbitrarily long raw audio wave as an input, and produces a 1024-dim feature vector. We append to that a fully connected layer to produce a 28-dim vector as the final output of the neural network. The first 14 dimensions are the one-hot encoding of primitive shapes and the next 10 dimensions are encodings of the specific modulus. The last 4 dimensions regress the initial height, the two Rayleigh damping coefficients and

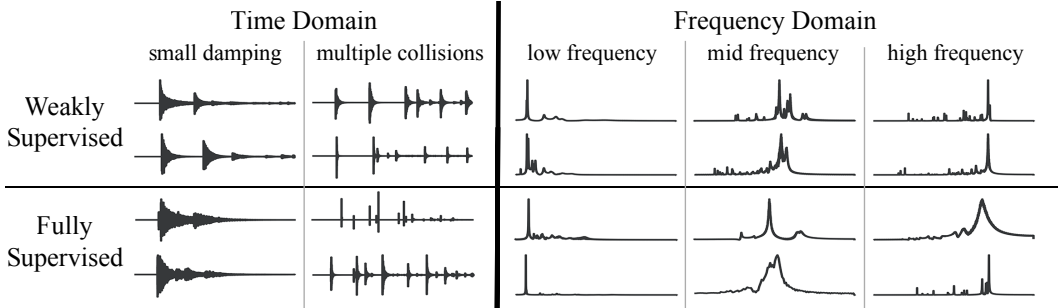

Figure 4: Visualization of top two sound waves that activate the hidden unit most significantly, in temporal and spectral domain. Their common characteristics can reflect the values of some latent variables, *e.g.* Rayleigh damping, restitution and specific modulus from left to right. Both weakly and fully supervised models capture similar features.

| Inference Model | | Latent Variables | | | | |
|---|---|---|---|---|---|---|
| | | shape | mod. | height | $\alpha$ | $\beta$ |
| Unsupervised | initial | 8% | 10% | 0.179 | 0.144 | 0.161 |
| | final | 54% | 56% | 0.003 | 0.069 | 0.173 |
| Self-supervised | initial | 14% | 16% | 0.060 | 0.092 | 0.096 |
| | final | 52% | 62% | 0.005 | 0.061 | 0.117 |
| Weakly supervised | initial | 18% | 12% | 0.018 | 0.077 | 0.095 |
| | final | 62% | 66% | 0.005 | 0.055 | 0.153 |
| Fully supervised | final | 98% | 100% | 0.001 | 0.001 | 0.051 |

Table 3: Initial and final classification accuracies (as percentages) and parameter MSE errors of three different inference models after 80 iterations of MCMC. Initial unsupervised numbers indicate chance performance. Results from the fully supervised model show performance bounds. $\alpha$ and $\beta$ are Rayleigh damping coefficients.

the restitution respectively. All the regression dimensions are normalized to a $[-1, 1]$ range. The weakly supervised model preserves the structure of the fully supervised one, but with an 8-dim final output: 3 for shape attributes, 1 for height, and 4 for materials. We used stochastic gradient descent for training, with a learning rate of 0.001, a momentum of 0.9 and a batch size of 16. Mean Square Error(MSE) loss is used for back-propagation. We implemented our framework in Torch7 [Collobert et al., 2011], and trained all models from scratch.

**Results**    Results for the four inference models proposed above are shown in Table 3. For shapes and specific modulus, we evaluate the results as classification accuracies; for height, Rayleigh damping coefficients, and restitution, results are evaluated as MSE. Before calculating MSE, we normalize values of each latent variable to $[-1, 1]$ interval, so that the MSE score is comparable across variables.

From Table 3, we can conclude that self-supervised and weakly supervised models benefit from the better initialization to the analysis-by-synthesis algorithm, especially on the last four continuous latent variables. One can also observe that final inference accuracies and MSEs are marginally better than for the unsupervised case. To illustrate the rate of convergence, we plot the likelihood value, $\exp(-kd)$ where $d$ is the distance of sound features, along iterations of MCMC in Figure 5. The mean curve of self-supervised model meets our expectation, *i.e.*, it converges much faster than the unsupervised model, and reaches a slightly higher likelihood at the end of 30 iterations. The fully supervised model, which is trained on 200,000 audios with the full set of ground-truth labels, yields near-perfect results for all latent variables.

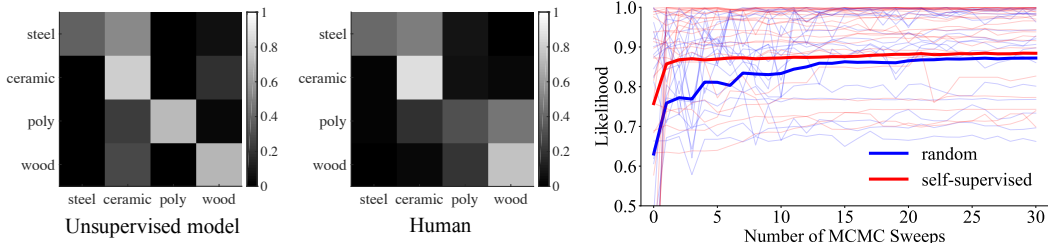

Figure 5: Left and middle: confusion matrix of material classification performed by human and our unsupervised model. Right: mean likelihood curve over MCMC iterations.

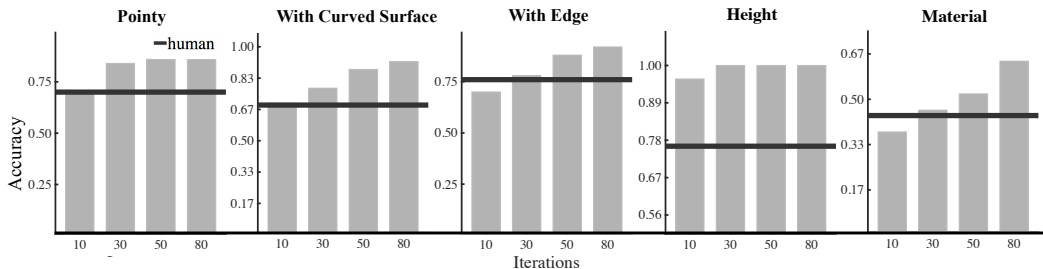

Figure 6: Human performance and unsupervised performance comparison. The horizontal line represents human performance for each task. Our algorithm closely matches human performance.

## 5 Evaluations

We first evaluate the performance of our inference procedure by comparing its performance with humans. The evaluation is conducted using synthetic audio with their ground truth labels. Then, we investigate whether our inference algorithm performs well on real-world recordings. Given recorded audio, our algorithm can distinguish the shape from a set of candidates.

### 5.1 Human Studies

We seek to evaluate our model relative to human performance. We designed three tasks for our subjects: inferring the object's shape, material and height-of-fall from the sound, intuitive attributes when hearing an object fall. Those tasks are designed to be classification problems, where the labels are in accordance with coarse labels used by our weakly-supervised model. The study was conducted on Amazon Mechanical Turk. For each experiment (shape, material, height), we randomly selected 52 test cases. Before answering test questions, the subject is shown 4 training examples with ground truth as familiarization of the setup. We collected 192 responses for the experiment on inferring shape, 566 for material, and 492 for height, resulting in a total of 1,250 responses.

**Inferring Shapes**  After becoming familiar with the experiment, participants are asked to make three binary judgments about the shape by listening to our synthesized audio clip. Prior examples are given for people to understand the distinctions of "with edge," "with curved surface," and "pointy" attributes. As shown in Figure 6, humans are relatively good at recognizing shape attributes from sound and are around the same level of competency when the unsupervised algorithm runs for 10∼30 iterations.

**Inferring Materials**  We sampled audio clips whose physical properties – density, Young's modulus and damping coefficients – are in the vicinity of true parameters of steel, ceramic, polystyrene and wood. Participants are required to choose one out of four possible materials. However, it can still be challenging to distinguish between materials, especially when sampled ones have similar damping and specific modulus. Our algorithm confuses steel with ceramic and ceramic with polystyrene occasionally, which is in accordance with human performance, as shown in Figure 5.

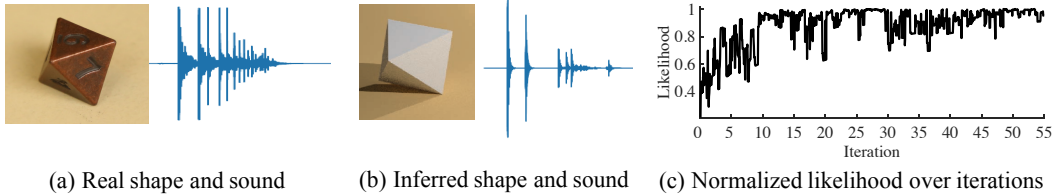

|(a) Real shape and sound|(b) Inferred shape and sound|(c) Normalized likelihood over iterations|

Figure 7: Results of inference on real world data. The test recording is made by dropping the metal dice in (a). Our inferred shape and reproduced sound is shown in (b). Likelihood over iteration is plotted in (c).

**Inferring Heights**    In this task, we ask participants to choose whether the object is dropped from a high position or a low one. We provided example videos and audios to help people anchor reference height. Under our scene setup, the touchdown times of the two extremes of the height range differ by 0.2s. To address the potential bias that algorithms may be better at exploiting falling time, we explicitly told humans that the silence at the beginning is informative. Second, we make sure that the anchoring example is always available during the test, which participants can always compare and refer to. Third, the participant has to play each test clip manually, and therefore has control over when the audio begins. Last, we tested on different object shapes. Because the time of first impact is shape-dependent, differently shaped objects dropped from the same height would have first impacts at different times, making it harder for the machine to exploit the cue.

## 5.2    Transferring to Real Scenes

In addition to the synthetic data, we designed real world experiments to test our unsupervised model. We select three candidate shapes: tetrahedron, octahedron, and dodecahedron. We record the sound a metal octahedron dropping on a table and used our unsupervised model to recover the latent variables. Because the real world scenarios may introduce highly complex factors that cannot be exactly modeled in our simulation, a more robust feature and a metric are needed. For every audio clip, we use its temporal energy distribution as the feature, which is derived from spectrogram. A window of 2,000 samples with a 1,500 sample overlap is used to calculate the energy distribution. Then, we use the earth mover's distance (EMD) [Rubner et al., 2000] as the metric, which is a natural choice for measuring distances between distributions.

The inference result is illustrated in Figure 7. Using the energy distribution with EMD distance measure, our generated sound aligns its energy at major collision events with the real audio, which greatly reduces ambiguities among the three candidate shapes. We also provide our normalized likelihood function overtime to show our sampling has converged to produce highly probable samples.

# 6    Conclusion

In this paper, we propose a novel model for estimating physical properties of objects from auditory inputs, by incorporating the feedback of an efficient audio synthesis engine in the loop. We demonstrate the possibility of accelerating inference with fast recognition models. We compare our model predictions with human responses on a variety of judgment tasks and demonstrate the correlation between human responses and model estimates. We also show that our model generalizes to some real data.

## Acknowledgements

The authors would like to thank Changxi Zheng, Eitan Grinspun, and Josh H. McDermott for helpful discussions. This work is supported by NSF #1212849 and #1447476, ONR MURI N00014-16-1-2007, Toyota Research Institute, Samsung, Shell, and the Center for Brain, Minds and Machines (NSF STC award CCF-1231216).

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
