[Reviews · NeurIPS 2017]

Reviewer 1



This paper contains a lot of ideas and methods packed into a tightly wound package - less would have been certainly more: The general aim is to build a system that can mimick the human ability to recognise shape and material objects from their sounds. Certainly a nice idea to explore within a NIPS community. To this end a generative model is defined, and a process for inference is proposed ("Physics-Based Audio Engine") that uses a physics simulation, this physics simulation is coupled (somehow) to a sound generation algorithm. This appears to use (somehow) the physics simulation engine representation of the vibrating surfaces of the objects to render the sounds. This engine is then used to create training stimuli for 4 variants of machine learning algorithms to learn appropriate representations. Again the descriptions are rather curt, so it is difficult to understand how the algorithms actually operate and how they perform. Into the paper mix we also have now human experiments that aims to compare the algorithms with that of the human observer. A description of how the experiment was conducted with human users is missing, however human users are almost always outperformed by the model. While it is up to this point unclear why the authors did not use real data when assessing their models, we are treated to real at the end (Figure 7) but without any meaningful information for us to understand how their method performs. In my opinion these are the major issues 1. Too much information therefore too superficially presented. 2. It is unclear how good the generative model is, comparison to real data would have been help here in anchoring it first. 3. The use of 4 different algorithms for classification is a nice effort, but it remains unclear to me what the point is (do the authors have a clear hypothesis?) 4. The human data is in itself interesting, but at this point it appears unclear how it was obtained, and how it informs us further on the work done in the paper.

Reviewer 2



This paper presents a system to infer shape and material of falling objects from sound. The main system follows the generative approach: its pipeline consists of a physics engine for simulating rigid body dynamics (given latent variables that characterize various features of the object) and an audio synthesis system. Four different inference models are explored in the paper ranging from an unsupervised scheme, in which a Gibbs sampler is used to infer the latent variables that give rise to a given test audio clip; to a fully-supervised model in which a oracle model is trained from the ground-truth data is obtained from the generative model. Finally, a study was conducted to compare the performance on object recognition between human and the inference model, showing that the latter performs comparably/better. PROS: - The first pages are relatively well written. Both the architecture and the experimental setups are well documented. - The system reaches (super) human-level performance in shape and material recognition from (synthetic) sounds. - The relation to previous literature is well documented. CONS: - Although the paper tackles a hard task using a sophisticated system, there is little to no significant ML contribution. - Many typos. Paper gets quite confusing towards the last pages and feels like having been written up in a rush. - There is no comparison to previous systems.

Reviewer 3



This paper describes an analysis-by-synthesis system for identifying properties of objects bouncing on a surface including material, shape, and height of fall. A thorough investigation is conducted on synthetic data using various constraints on the training of the model, showing that it is accurately able to recover these basic facts from new recordings. An experiment on real data comparing human listeners with the system shows that it is comparable and may outperform the humans in the height task. The system uses a surprisingly simple comparison method between the observed and synthesized signals, but it appears to work well. The literature review is short, but quite good. The paper has several minor weaknesses: * It is not clear if the ability of the model to detect fall height is because of the absolute timing of the simulations. Falling from a greater height leads to a longer delay before the first impact. This is obvious to an algorithm analyzing fixed-sized wav files, but not to a human listening to sound files with somewhat unknown silent beginnings. A fairer comparison would be to add a random amount of delay before starting the sounds for both listeners. * The comparison method is changed between the synthetic and real tasks, which seems unfair. If it is necessary to use a more complex comparison method for the real task, then also use it for the synthetic one. * Line 226 reports several analysis parameters in samples, but never states the sample rate. Please describe these quantities in seconds or ms or provide the sample rate so the reader can perform the conversion themselves. Overall, this is a strong paper that has gotten a relatively old and appealing idea to work much better than in the past.